# Antithrombotic Strategies in Patients with Atrial Fibrillation Following Percutaneous Coronary Intervention: A Systemic Review and Network Meta-Analysis of Randomized Controlled Trials

**DOI:** 10.3390/jcm9041062

**Published:** 2020-04-08

**Authors:** Su-Kiat Chua, Lung-Ching Chen, Kou-Gi Shyu, Jun-Jack Cheng, Huei-Fong Hung, Chiung-Zuan Chiu, Chiu-Mei Lin

**Affiliations:** 1School of Medicine, Fu Jen Catholic University, New Taipei City 242, Taiwan; M006507@ms.skh.org.tw; 2Division of Cardiology, Department of Internal Medicine, Shin Kong Wu Ho-Su Memorial Hospital, Taipei 111, Taiwan; 3Department of Internal Medicine, Shin Kong Wu Ho-Su Memorial Hospital, Taipei 111, Taiwan; 4Department of Emergency Medicine, Shin Kong Wu Ho-Su Memorial Hospital, Taipei 111, Taiwan

**Keywords:** atrial fibrillation, antithrombotic therapy, percutaneous coronary intervention, dual anti-thrombotic therapy, triple antithrombotic therapy

## Abstract

Up to 10% of patients with atrial fibrillation (AF) undergo percutaneous coronary intervention (PCI). A systematic review and network meta-analysis were conducted by searching PubMed, Embase, and the Cochrane database of systematic reviews for randomized control trials that studied the safety and efficacy of different antithrombotic strategies in these patients. Six studies, including 12,158 patients were included. Compared to that in the triple antithrombotic therapy group (vitamin K antagonist (VKA) plus P2Y_12_ inhibitor and aspirin), thrombolysis in myocardial infarction (TIMI) major bleeding was significantly reduced in the dual antithrombotic therapy (non-vitamin K oral anticoagulants (NOACs) plus P2Y_12_ inhibitor) group by 47% (Odds ratio (OR), 0.53; 95% credible interval [CrI], 0.35–0.78; I^2^ = 0%). Besides, NOAC plus a P2Y_12_ inhibitor was associated with less intracranial hemorrhage compared to VKA plus single antiplatelet therapy (OR: 0.20, 95% CrI: 0.05–0.77). There was no significant difference in the trial-defined major adverse cardiac events or the individual outcomes of all-cause mortality, cardiovascular death, myocardial infarction, stroke or stent thrombosis among all antithrombotic strategies. In conclusion, antithrombotic strategy of NOACs plus P2Y_12_ inhibitor is safer than, and as effective as, the strategies including aspirin when used in AF patients undergoing PCI.

## 1. Introduction

Up to 30% of patients with atrial fibrillation (AF) have concomitant coronary artery disease (CAD) and approximately 10% of them undergo percutaneous coronary intervention (PCI) [1]. For physicians managing AF and concomitant ischemic heart disease, particularly acute coronary syndrome (ACS) following PCI, it is challenging to determine which antithrombotic strategies provide the best balance of safety and efficacy [2]. Vitamin K antagonists (VKAs) or non-vitamin K oral anticoagulants (NOACs) are necessary to prevent stroke in patients with AF [2,3], whereas dual antiplatelet therapy (DAPT) with P2Y_12_ inhibitors and aspirin is required to prevent stent thrombosis in patients undergoing PCI [4,5]. Current guidelines for AF patients undergoing PCI recommend using triple thrombotic therapy, which is a combination of VKA or NOAC with DAPT to reduce the risk of cardioembolic and coronary thrombotic complications [4,5]. However, triple thrombotic therapy substantially increases the risk of bleeding complications [6,7].

In recent years, a number of randomized controlled trials (RCTs) have attempted to compare the safety and efficacy of different antithrombotic strategies after PCI for stable CAD or ACS in patients with AF [8,9,10,11,12]. In addition, several meta-analyses have concluded that administering dual antithrombotic therapy with NOAC and P2Y_12_ inhibitors results in fewer major bleeding events, but similar efficacy compared to triple therapy, including VKA or NOAC with DAPT [13,14,15]. However, most of these meta-analyses did not include the latest reported edoxaban-based antithrombotic regimen in patients with AF following the successful percutaneous coronary intervention (ENTRUST-AF PCI) trial, which was found to be insufficient to detect differences in safety outcome between the antithrombotic strategies with and without aspirin in such patients [16]. Furthermore, these previous studies are complemented by a similar risk of thromboembolic events and major cardiovascular events (MACEs) between dual and triple antithrombotic therapy. Moreover, it is not clear how different classes of oral anticoagulant (OAC), such as VKA or NOAC, will affect the safety and efficacy outcomes in AF patients undergoing PCI. To address the above issues and reduce the selection bias by observational studies, we conducted a Bayesian network meta-analysis of RCTs with the goal of analyzing antithrombotic strategies in this high-risk population.

In this network meta-analysis, simultaneous comparisons of VKA or NOAC with single versus dual antiplatelet therapy were analyzed for safety and efficacy outcomes in patients with AF undergoing PCI. In addition, we compared our findings with previous meta-analyses that compared dual and triple antithrombotic therapy in AF patients undergoing PCI.

## 2. Method

### 2.1. Study Selection, Search Strategy and Outcome Measures

This systematic review and network meta-analysis adhered to the Preferred Reporting Items for Systematic Reviews and Meta-Analysis (PRISMA) guidelines [17]. The inclusion criteria for this network meta-analysis were as follow: (1) all relevant Phase 3 RCTs comparing dual antithrombotic therapy (defined as VKA or NOAC with single antiplatelet agent [SAPT]) versus triple antithrombotic therapy (defined as VKA or NOAC with DAPT) in patients with AF undergoing PCI; (2) reported major and minor bleeding using thrombolysis in myocardial infarction (TIMI), the Global Utilization of Streptokinase and TPA for Occluded Arteries definition for bleeding (GUSTO), or the Bleeding Academic Research Consortium (BARC) classification; (3) reported major adverse cardiovascular events (MACE), including all-cause death, cardiovascular death, myocardial infarction, stroke, and stent thrombosis; and (4) at least 6 months follow-up period. The primary exclusion criteria were observational studies, registry data, editorials, case series, crossover trials, duplicate studies, and non-original data.

We searched PubMed, Embase, and the Cochrane database through December of 2019 using the following key words in various combinations: “percutaneous coronary intervention”, “acute coronary syndrome”, “coronary angioplasty”, “coronary stenting”, “triple antithrombotic therapy”, “dual antithrombotic therapy”, “anticoagulant”, “antiplatelet”, “vitamin K antagonists”, “warfarin”, “dabigatran”, “rivaroxaban”, “apixaban”, “edoxaban”, “aspirin”, “clopidogrel”, “atrial fibrillation”, and “randomized clinical trial”. Two investigators (SKC, JJC) independently reviewed the titles or abstracts of the studies to determine whether they met the inclusion and exclusion criteria. Disagreements were resolved via consensus and by a third investigator (LCC).

We extracted information from the enrolled RCTs to evaluate the following outcomes: (1) TIMI major bleeding, (2) TIMI minor bleeding, (3) trial-defined safety outcome, (4) intracranial hemorrhage, (5) trial-defined MACE, (6) all-cause mortality, (7) cardiovascular death, (8) myocardial infarction (MI), (9) stroke, and (10) stent thrombosis.

### 2.2. Statistical Analysis and Risk of Bias

We used the number of patients randomized for the intention-to-treat sample size. A Bayesian network random effect meta-analysis was used to compare the effects of four antithrombotic strategies (VKA and DAPT, VKA and SAPT, NOAC and DAPT, and NOAC and P2Y_12_ inhibitors) for each outcome. The pooled odds ratio (ORs) and their 95% credible intervals (95% CrI) were estimated for both direct and indirect comparisons. Heterogeneity across studies was also assessed using the Cochrane Q test and Higgins I^2^ statistics. The Higgins I^2^ statistics was used to determine the degree heterogeneity (I^2^ <25%—low, 25–50%—moderate, and >50%—high degree of heterogeneity) among the enrolled studies. The Markov Chain Monte Carlo (MCMC) method was utilized to obtain estimates of posterior distribution. The MCMC model was repeated 50,000 times to allow for convergence, and 20,000 sample burn-in was performed to produce the probability statements for each analysis. Convergence of iterations was evaluated using Gelman–Rubin–Brooke statistic and trace plots. The diagnostics for the network meta-analysis models under the assumption of evidence consistency for the above 10 outcomes showed that the four MCMC chains mixed well regardless of their different starting points. Additionally, trace plots showed that the MCMC chains converged well (Appendix A). All statistical analyses were performed with R (V.3.6.1) and the GeMTC (V.0.8-2) package, along with the Markov Chain Monte Carlo engine JAGS (V.3.4.0).

## 3. Results

### 3.1. Enrollment of Studies

In total, we identified 305 studies. Of these, 200 were deemed irrelevant after title and abstract screening, and 105 were assessed for eligibility using the full text (Figure 1). Finally, six studies met the inclusion criteria [8,9,10,11,12,16]. Of these, the ISAR-TRIPLE trial had a different study design compared to the others in that both arms of the study participants were treated with the same triple therapy for the first six weeks, after which, the dual therapy group received VKA and aspirin, while the triple therapy group received VKA and DAPT. Therefore, to ensure valid comparisons, only the event data from the landmark analysis of the ISAR-TRIPLE trial was analyzed in our network meta-analysis. All the enrolled patients in WOEST and ISAR-TRIPLE studies used clopidogrel plus aspirin as DAPT, wherein PIONEER, RE-DUAL PCI, AGUSTUS, ENTRUST-AF PCI, more than 90% of the patients received clopidogrel plus aspirin, only less than 10% received ticagrelor or prasugrel [8,9,10,11,12,16].

The trial design, treatment strategies, and safety and efficacy outcomes of the six included RCTs are summarized in Appendix A. All RCTs were judged to be at a low risk of bias.

### 3.2. Structure of the Network Meta-Analysis

The network of antithrombotic strategies used in the main analysis is summarized in Figure 2. Four antithrombotic strategies, VKA with DAPT, VKA with aspirin or a P2Y_12_ inhibitor, NOAC with DAPT, and NOAC with a P2Y_12_ inhibitor, were compared in our network meta-analysis. For a safety and efficacy analysis, we compared four NOACs, including rivaroxaban, dabigatran, apixaban, and edoxaban in this network meta-analysis.

### 3.3. Network Meta-Analysis Safety Outcomes

Regarding the safety outcomes, using NOAC with a P2Y_12_ inhibitor was associated with less TIMI major bleeding (OR: 0.53, 95% CrI: 0.35–0.78), TIMI major and minor bleeding (OR: 0.52, 95% CrI: 0.30–0.89), trial-defined safety outcome (OR: 0.54, 95% CrI: 0.33–0.86) and intracranial hemorrhage (OR: 0.31, 95% CrI: 0.12–0.73) compared to using VKA with DAPT (Figure 3). In addition, VKA with a P2Y_12_ inhibitor was associated with lower trial-defined safety outcomes than VKA with DAPT (OR: 0.47, 95% CrI: 0.22–0.94). Pairwise comparisons among the strategies are summarized in Appendix A. The use of NOAC with a P2Y_12_ inhibitor was associated with favorable any bleeding outcome when compared to other antithrombotic strategies. Moreover, the use of NOAC with a P2Y_12_ inhibitor was associated with less intracranial hemorrhage compared to VKA with SAPT (OR: 0.20, 95% CrI: 0.05–0.78).

### 3.4. Network Meta-Analysis for Efficacy Outcomes

Overall, there were no significant differences for trial-defined primary MACEs among the four antithrombotic strategies. Moreover, the four antithrombotic strategies had other comparable efficacy outcomes with respect to all-cause death, cardiovascular death, stroke, MI, and stent thrombosis (Figure 4). Pairwise comparisons regarding efficacy outcomes among the strategies are summarized in Appendix A. The odds of any efficacy outcomes were the same irrespective of the antithrombotic strategy used.

The performance of the four antithrombotic strategies is summarized in a two-dimensional forest plot of ORs (Figure 5). Compared to VKA with DAPT, NOAC with a P2Y_12_ inhibitor was associated with fewer TIMI major bleeding with no significant difference in the rate of primary MACEs outcome. 

## 4. Discussion

This network meta-analysis of RCTs is the first meta-analysis to include four NOACs and had demonstrated several important findings. We found that dual antithrombotic agents with NOAC and P2Y_12_ inhibitors were associated with a lower rate of bleeding, including major or minor bleeding, and study defined primary safety endpoint compared to triple therapy that included aspirin. We also found that there was no significant difference in MACE. These results support the prescription of NOAC with P2Y_12_ inhibitors as the favorable regimen in AF patients undergoing PCI. Routine use of triple therapy in such patients should generally be avoided.

### 4.1. Dual Versus Triple Antithrombotic Therapy

Balancing the risk of bleeding and thromboembolism with dual or triple antithrombotic therapy in AF patients undergoing PCI requires patient-by-patient evaluation. Current guidelines recommend a short course of triple antithrombotic therapy, including OAC and DAPT, in patients with OAC indication who are undergoing PCI [18]. Several meta-analyses comparing the safety and efficacy of dual and triple antithrombotic therapies in these high risk patients are summarized in Table 1 [13,14,15,19,20,21,22,23,24,25,26]. The network meta-analysis performed by Gong et al. showed that the combination of VKA with single antiplatelet therapy (SAPT) yielded the best safety and efficacy, with a very low dose of rivaroxaban and DAPT deemed to be an acceptable alternative to standard triple therapy with VKA and DAPT [21]. Recently, two meta-analyses of RCTs demonstrated that dual antithrombotic therapy with NOAC and P2Y_12_ inhibitors was superior regarding bleeding, and equivalent regarding efficacy, compared to VKA and DAPT [13,14]. Our network meta-analysis corroborates the finding that dual therapy with an NOAC and P2Y_12_ inhibitor superior to triple therapy in AF patients undergoing PCI; This conclusion is based on the observed 46% reduction in major or minor TIMI bleeding with similar MACE outcomes. Furthermore, the use of a NOAC and P2Y_12_ inhibitor was associated with a 69% reduction in intracranial hemorrhage, which is one of the most severe complications of standard triple therapy. The precise reasons for the decreased bleeding but similar efficacy of dual therapy compared to triple therapy are unknown, but there are several possibilities. First, new generation drug-eluting stents with a low risk of stent thrombosis may contribute to this effect by decreasing the necessity of DAPT in these patients [27]. Second, P2Y_12_ inhibitors, particularly clopidogrel, are more potent platelet inhibitor than aspirin and have less bleeding risk [28].

### 4.2. Dual Antithrombotic Therapy with NOAC or VKA and P2Y_12_ Inhibitors

Most of the meta-analyses regarding VKA and P2Y_12_ inhibitors with and without aspirin have demonstrated that dual therapy is associated with a similar risk of bleeding complications and a similar risk of MACEs compared to triple therapy. However, the study by Zhu et al. found that dual therapy had a significantly reduced (40% lower) bleeding risk compared to triple therapy [20]. These data suggest that dual therapy with VKA and P2Y_12_ inhibitors is more favored than standard triple therapy with VKA, P2Y_12_ inhibitors, and aspirin, and that standard triple therapy should be avoided in such patients. Although the ENTRUST-AF PCI trial was found to be underpowered to detect differences in bleeding risk between dual and triple thrombotic therapy in such patients [16], the other three NOAC trials demonstrated a significant reduction in bleeding complications with dual therapy compared to triple therapy, with comparable efficacy outcomes [10,11,12]. In addition, several meta-analyses, including these NOAC trials, demonstrated that NOACs and P2Y_12_ inhibitors resulted in a 40–50% risk reduction in bleeding complications compared to standard triple therapy [13,14,15,26]. Our network meta-analysis also confirmed that dual therapy with NOACs and P2Y_12_ inhibitors had favorable safety outcomes compared to dual therapy with VKA and P2Y_12_ inhibitors.

### 4.3. Previous and Present Meta-Analyses

Previous meta-analyses regarding the use of dual and triple antithrombotic therapy in AF patients undergoing PCI are summarized in Table 1. Results from these meta-analyses have conflicting results because of the difference of included studies and definition of safety outcome, such as major bleeding in each meta-analysis. The present analysis was different in many aspects. First, the present network meta-analysis restricted to data from high-quality RCTs to avoid possible bias. Second, the present study included trials in which all patients received PCI for ACS or stable CAD, while the previous meta-analyses included trials that enrolled patients with and without undergoing PCI. Third, previous meta-analyses included patients receive OAC indicated for AF, mechanical valve, or systemic thrombotic event while more than 95% of the population in the present meta-analysis received OAC because of AF. Finally, this is the first network meta-analysis to include four different NOACs, which increases the relevance of our meta-analysis.

### 4.4. Clinical Implication

In an era when physicians must balance bleeding risk and ischemic events risk in AF patients undergoing PCI, our network meta-analysis demonstrates that dual therapy with NOACs and P2Y_12_ inhibitors is better than conventional triple therapy for bleeding outcomes, and comparable in terms of efficacy outcomes. These results have significant clinical implications, as bleeding is associated with the interruption of antithrombotic therapy, which subsequently increases the risk of MACE in standard triple therapy [30]. These findings are crucial for the management of patients at high risk of bleeding in whom standard triple therapy should be avoid.

## 5. Limitation

This network meta-analysis has several limitations. First, there was heterogeneity between the included studies; in factors such as study design, indication of PCI (ACS or non-ACS), CHA_2_DS_2_-VASc or HAS-BLED scores, type of stent (drug-eluting or bare metal stent), type, dose, and duration of antithrombotic therapy, and duration of follow-up, which may potentially affect the interpretation of the results. Second, apart from the ISAR-TRIPLE trial, which used aspirin as the antiplatelet therapy, the other five trials used clopidogrel as a P2Y_12_ inhibitor in most of their dual therapy strategies. Further analysis should be conducted on more potent P2Y_12_ inhibitors, such as ticagrelor and prasugrel, which have a similar safety and efficacy in combination with OAC. Third, approximately 70% and 80% of enrolled patients in the WOEST and ISAR-TRIPLE trials, respectively. However, subgroup analysis of the WOEST and ISAR-TRIPLE trials showed no statistical differences in primary outcomes based on OAC indication, including AF, mechanical valves, and venous thrombosis.

## 6. Conclusions

Our systemic review and network meta-analysis demonstrates that NOAC and P2Y_12_ inhibitors may be a better choice than standard triple therapy or dual therapy with OAC and P2Y_12_ inhibitors in AF patients undergoing PCI.

## Figures and Tables

**Figure 1 jcm-09-01062-f001:**
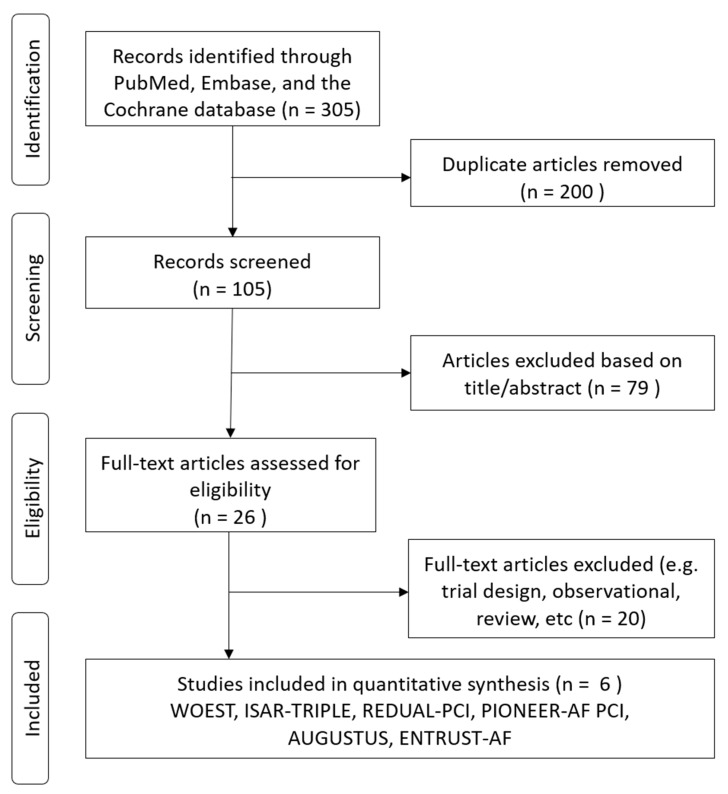
Flow diagram of included studies.

**Figure 2 jcm-09-01062-f002:**
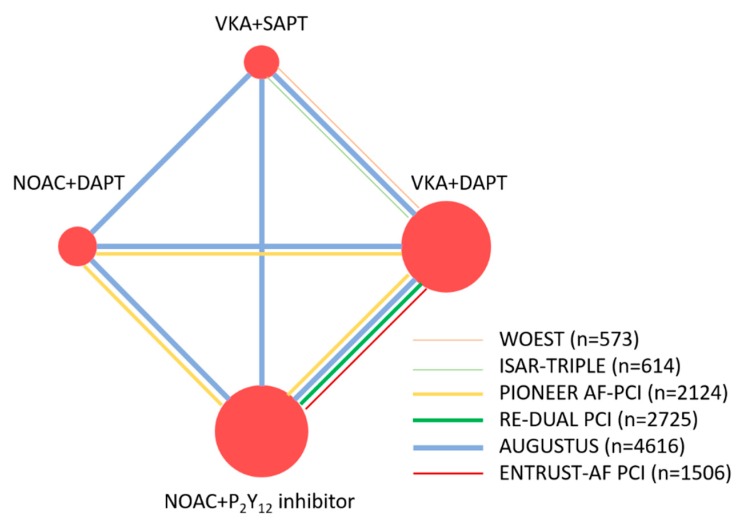
Network of four antithrombotic therapy strategies. The nodes represent the antithrombotic therapy strategies to be compared; the size of nodes is proportional to the number of patients included for the different antithrombotic therapy strategies. The edges represent the direct comparisons between the antithrombotic therapy strategies and the thickness is proportional to the sample size of the studies.

**Figure 3 jcm-09-01062-f003:**
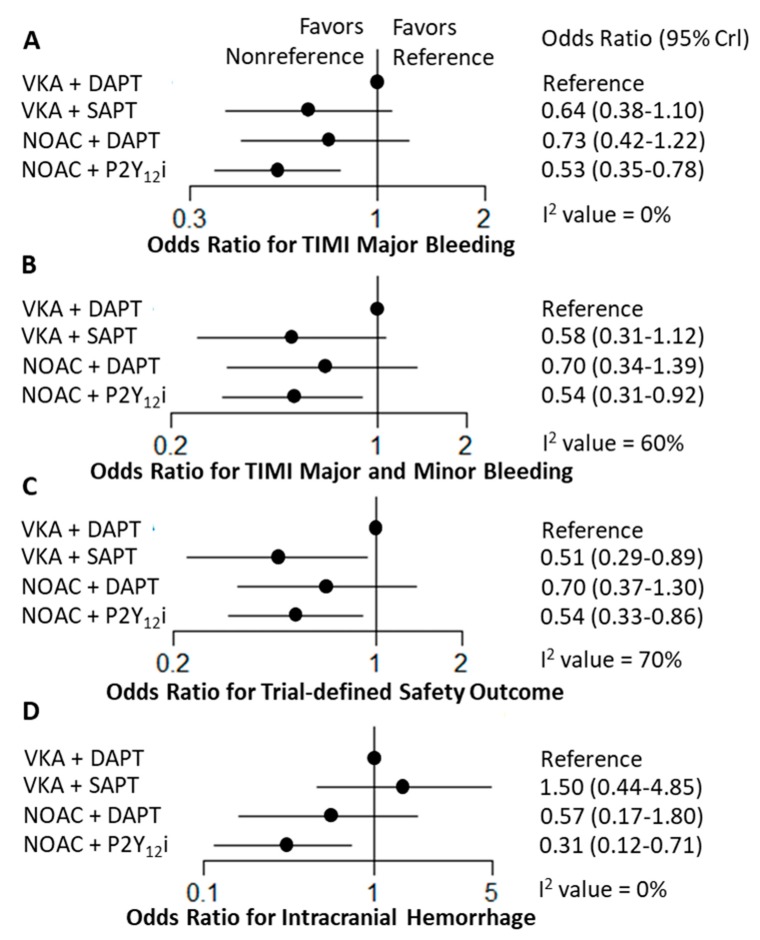
Forest plots for safety outcomes. Odds ratio and 95% credible intervals (CrI) compared with vitamin antagonist (VKA) plus dual antiplatelet therapy (DAPT) are plotted. The estimated between-trial effect heterogeneity and its 95% CrI (in standard deviation of the log odds ratio scale) from network meta-analysis for each outcome is (**A**) TIMI major bleeding, 0.22 (95% CrI, 0.10–0.55), I^2^ = 0%; (**B**) TIMI major and minor bleeding, 0.46 (95% CrI, 0.09–0.84), I^2^ = 60%; (**C**) Trial-defined primary safety outcome, 0.48 (95% CrI, 0.08–1.08), I^2^ = 70%; (**D**) Intracranial hemorrhage, 0.41 (95% CrI, 0.26–1.08), I^2^ = 0%. DAPT, dual antiplatelet therapy; NOAC, non-vitamin K antagonist oral anticoagulant; SAPT, single antiplatelet therapy; VKA, vitamin K antagonist.

**Figure 4 jcm-09-01062-f004:**
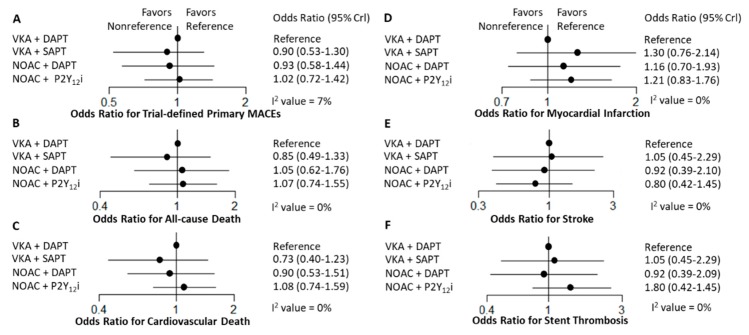
Forest plots for efficacy outcomes. Odds ratio and 95% credible intervals (CrI) compared with vitamin antagonist (VKA) plus dual antiplatelet therapy (DAPT) are plotted. The estimated between-trial effect heterogeneity and its 95% CrI (in standard deviation of the log odds ratio scale) from network meta-analysis for each outcome is (**A**) Trial-defined primary MACEs, 0.23 (95% CrI, 0.09–0.55), I^2^ = 7%; (**B**) All-cause death, 0.24 (95% CrI, 0.13–0.60), I^2^ = 0%; (**C**) Cardiovascular death, 0.17 (95% CrI, 0.12–0.46), I^2^ = 0%; (**D**) Myocardial infarction, 0.15 (95% CrI, 0.05–0.35), I^2^ = 0%. (**E**) Stroke, 0.41 (95% CrI, 0.06–0.90), I^2^ = 0%; (**F**) Stent thrombosis, 0.30 (95% CrI, 0.09–0.69), I^2^ = 0%. MACEs: major adverse cardiovascular events; DAPT: dual antiplatelet therapy; NOAC: non-vitamin K antagonist oral anticoagulant; SAPT: single antiplatelet therapy; VKA: vitamin K antagonist.

**Figure 5 jcm-09-01062-f005:**
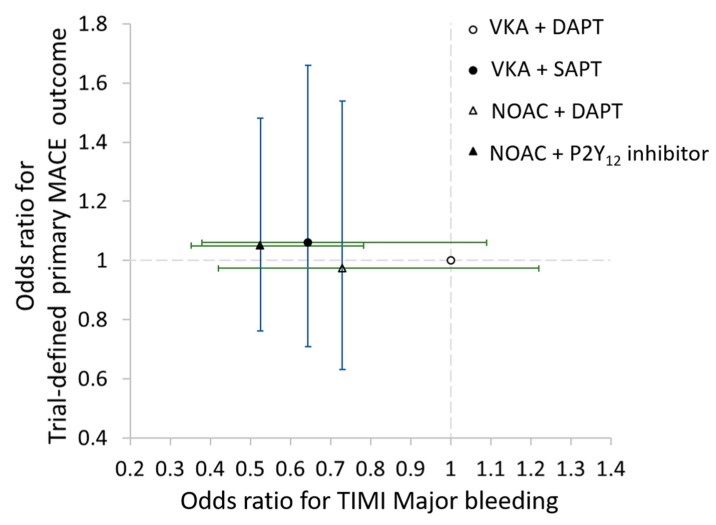
Odds ratio (OR) for TIMI major bleeding and the trial defined major adverse cardiovascular events (MACE).

**Table 1 jcm-09-01062-t001:** Comparisons of previous and present meta-analyses related to the combination of anticoagulant and P2Y_12_ inhibitors with and without aspirin in atrial fibrillation patients after percutaneous coronary intervention.

Author, Year	Target Population	Number of Included Studies	Timespan of All Studies	Total Number of Patients	Safety Outcome (Major Bleeding)	Efficacy Outcome
**Network Meta-Analysis of RCTs**						
Gong et al., 2017 [21]	Patients with AF undergoing PCI	7 P, 5 R, 3 RCTs (WOEST, PIONEER -AF PCI, ROCKET AF post-hoc)	2008 to 2016	13,104	DT vs. TT Risk Ratio = 0.97 (95% CI: 0.29–3.35)	MACE: DT vs. TT Risk Ratio = 0.68 (95% CI: 0.43–0.98)
Bunmark et al., 2018 [26]	Patients with OAC undergoing PCI	4 RCTs (WOEST, PIONEER-AF PCI, REDUAL-PCI), 12 P, 14 R	2007 to 2017	22,179	DT vs. TT RR = 0.68 (95% CI: 0.49–0.94)	All-cause death: DT vs. TT RR = 0.40 (95% CI: 0.17–0.93)
Lopes et al., 2019 [13]	Patients with AF undergoing PCI	5 RCTs (WOEST, ISAR-TRIPLE, PIONEER AF-PCI, RE-DUAL PCI, AGUSTUS)	2013 to 2018	10,026	DT vs. TT OR = 0.49 (95% CI: 0.30–0.82)	MACE: DT vs. TT OR = 1.02 (95% CI: 0.71–1.47) All-cause death: DT vs. TT OR = 1.02 (95% CI: 0.59–1.74) Stroke: DT vs. TT OR = 0.77 (95% CI: 0.34–1.67)
Present study, 2020	Patients with AF undergoing PCI	6 RCTs (WOEST, ISAR-TRIPLE, PIONEER AF-PCI, RE-DUAL PCI, AGUSTUS, ENTRUST-AF PCI)	2013 to 2019	11,532	DT vs. TT HR = 0.53 (95% CI: 0.35–078)	MACE: DT vs. TT OR = 1.02 (95% CI: 0.72–1.42) All-cause death: DT vs. TT OR = 1.08 (95% CI: 0.72–1.60) Stroke: DT vs. TT: 0.80 (95% CI: 0.41–1.48)
**Systemic Review and Meta-Analysis**						
Chen et al., 2017 [19]	Patients with OAC undergoing PCI	2 RCTs (WOEST, ISAR-TRIPLE), 5 P, 5 R	2007 to 2016	30,823	TT vs. DT RR = 0.86 (95% CI: 0.74–0.99)	MACE: TT vs. DT RR = 0.82 (95% CI: 0.58–1.17) All-cause death: TT vs. DT RR = 0.90 (95% CI: 0.54–1.51) Stroke: TT vs. DT RR = 1.08 (95% CI: 0.56–2.07)
Zhu et al., 2017 [20]	Patients with AF and ischemic heart disease	8 P, 9 R	2010 to 2017	38,099	TT vs. DT RR = 1.65 (95% CI: 1.23–2.21)	MACE: TT vs. DT RR = 1.14 (95% CI: 0.75–1.73, p = 0.55) All-cause death: TT vs. DT RR = 1.21 (95% CI: 0.78–1.88) TE: TT vs. DT RR: 1.55 (95% CI: 0.89–2.72; p = 0.12)
Agarwal et al., 2017 [22]	Patients with OAC undergoing PCI	2 RCTs (WOEST, PIONEER-AF PCI), 6 P, 3 R	2007 to 2016	7276	TT vs. DT RR = 1.54 (95% CI: 1.20 to 1.98)	MACE: TT vs. DT RR = 1.03 (95% CI: 0.90 to 1.32) All-cause death: TT vs. DR RR = 0.98 (95% CI: 0.68 to 1.43) TE: TT vs. DR RR = 1.02 (95% CI: 0.49 to 2.10)
Yu et al., 2017 [23]	Patients with OAC undergoing PCI	3 RCTs (WOEST, ISAR-TRIPLE, PIONEER AF-PCI), 5 P, 6 R	2000 to 2016	32,825	TT vs. DT OR = 1.56 (95% CI: 0.98–2.49);	MACE: TT vs. DT OR = 0.97 (95% CI: 0.68 to 1.387) All-cause death: TT vs. DT OR = 2.11 (95% CI: 1.10–4.06) SE: TT vs. DT OR = 0.43 (95% CI: 0.30–0.62)
Cavallari et al., 2018 [24]	Patients with AF undergoing PCI	4 RCTs (WOEST, ISAR-TRIPLE, PIONEER AF-PCI, RE-DUAL PCI)	2013 to 2017	6036	DT vs. TT OR = 0.55 (95% CI: 0.39 to 0.78)	All-cause death: DT vs. TT OR = 0.81 (95% CI: 0.50 to 1.29) Stroke: DT vs. TT OR = 0.95 (95% CI: 0.58 to 1.57)
Golwala et al., 2018 [14]	Patients with AF undergoing PCI	4 RCTs (WOEST, ISAR-TRIPLE, PIONEER AF-PCI, RE-DUAL PCI)	2013 to 2017	5317	DT vs. TT HR = 0.53 95% CI: 0.36–0.85)	MACE: DT vs. TT HR = 0.85 (95% CI: 0.48–1.29) All-cause death: DT vs TT HR = 0.85 (95% CI: 0.46–1.37) Stroke: DT vs. TT HR = 0.94 (95% CI: 0.45–1.84)
Brunetti et al., 2018 [15]	Patients with AF undergoing PCI	2 RCTs (PIONEER AF-PCI, RE-DUAL PC)	2016 to 2017	4849	DT vs. TT RR = 0.59 (95% CI: 0.47–0.73)	MACE: DT vs. TT RR = 1.03 (95% CI: 0.89–1.19)
Liu et al., 2018 [25]	Patients with AF undergoing PCI	5 P, 9 R	2010 to 2016	11,697	TT vs. DT OR = 1.55 (95% CI: 1.16–2.09)	MACE: TT vs. DT OR = 0.97 (95% CI: 0.87–1.07) All-cause death: TT vs. DT OR = 0.92 (95% CI: 0.83–1.03) Stroke: TT vs. DT OR = 0.74 (95% CI: 0.59–0.93)
Gargiulo et al., 2019 [29]	Patients with AF undergoing PCI	4 RCTs (PIONEER AF-PCI, RE-DUAL PCI, AUGUSTUS, ENTRUST-AF PCI)	2016 to 2019	10,234	DT vs. TT RR = 0,66 (95% CI: 0.56-0.78)	MACE: DT vs. TT OR = 1.08 (95% CI: 0.95–1.23) All-cause death: DT vs. TT OR = 1.10 (95% CI: 0.91–1.34) Stroke: DT vs. TT OR = 1.00 (95% CI: 0.69–1.45)

CI: Confidence interval, DT: Dual therapy with OAC and single antiplatelet therapy, MACE: Major adverse cardiovascular events, OR: Odd ratios, P: Prospective studies, R: Retrospective studies, RCTs: Randomized controlled trials, RR: Relative risk. TT: triple therapy with OAC and dual antiplatelet therapy.

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
