# Peer review of "Antithrombotic Strategies in Patients with Atrial Fibrillation Following Percutaneous Coronary Intervention: A Systemic Review and Network Meta-Analysis of Randomized Controlled Trials"

_jcm, 2020, doi:10.3390/jcm9041062_

Round 1

Reviewer 1 Report

This network meta-analysis is a new ring in the chain of multiple previous meta-analyses to compare VKA and NOACs with single or dual antiplatelet therapy after PCI in patients with AF. Most of the previous meta-analyses have not included the most recent ENTRUST-AF PCI trial, which was, however, underpowered, and open-label design also caused problems in the interpretations. ISAR-TRIPLE data carries the problem that SAPT period started long after PCI diminishing the bleeding risks.

The analyses seem adequate to my eyes. The report is generally well-written and the limitations are acknowledged.

Author Response

This network meta-analysis is a new ring in the chain of multiple previous meta-analyses to compare VKA and NOACs with single or dual antiplatelet therapy after PCI in patients with AF. Most of the previous meta-analyses have not included the most recent ENTRUST-AF PCI trial, which was, however, underpowered, and open-label design also caused problems in the interpretations. ISAR-TRIPLE data carries the problem that SAPT period started long after PCI diminishing the bleeding risks.

The analyses seem adequate to my eyes. The report is generally well-written and the limitations are acknowledged.

Ans. Thank you for your comment. 

Reviewer 2 Report

It is an interesting Metanalysis, focusing on specific subject. It needs bit of refining in the language and expressions used, eg line 148.

Most importantly Figure 4 needs to be further explained.

Otherwise is quite concrete manuscript. 

Author Response

Response to reviewer 2

  1. It is an interesting Metanalysis, focusing on specific subject. It needs bit of refining in the language and expressions used, eg line 148.

Ans. Thank you for your comment. The revised article was edited by professional English editing service. 

  1. Most importantly Figure 4 needs to be further explained.

Ans. Thank you for your suggestion. Following other reviewer's suggestions, Figure 4 was shifted to supplementary Figure 2. We also have revised the description of this Figure to improve clarity.

Reviewer 3 Report

In this systemic review and network meta-analysis of six randomized controlled trials, the authors compared the safety and efficacy of antagonists (VKAs) or non-vitamin K oral anticoagulants (NOACs) with single or dual antiplatelet therapy in patients with atrial fibrillation who underwent percutaneous coronary intervention. They showed that in these patients, the combination of NOACs and P2Y12 inhibitor was safer than and as effective as triple therapy, including VKAs or NOACs with dual antiplatelet therapy. The authors should be congratulated for their huge amount of work. However, I have some concerns which need to be discussed. Please consider the following comments.

1.The abstract is not clear, especially the significance of terms dual “antithrombotic therapy” and “triple antithrombotic therapy”. It should be rewritten as clear as the end of the introduction of the manuscript. Moreover, the major adverse cardiac events are not defined.

2.It is important for readers to clearly detail what DAPT means in the different studies. Did all the studies consider the same drugs for DAPT?

3.It would have been interesting to perform subgroups analyses, according to age, renal failure, and other comorbidities.

4.You detailed in the statistical analysis section how you determined the heterogeneity across studied and how you performed sensitivity analyses. However, you did not provide any results. Please clarify this point.

5.Please add in figure3 for the different endpoints, the individual results of each study you considered, as well as the weight of the different studies included in each analysis and the value of I² for each endpoint.

6.There are too many figures. Some of them are not very informative and should be considered as supplemental data only. In particular, the Figure 4 is not so easy to understand for readers. Please clarify.

7.In the discussion, you should more focus on the added value of your results. You should also more discuss your results in the light of the existing literature.

8.There are too many abbreviations in the manuscript making it difficult to read. Same comment for tables.

9.Please correct all the spelling errors throughout the manuscript. An English editing by a narrative speaker would be appreciated.

Author Response

Response to reviewer 3

  1. The abstract is not clear, especially the significance of terms dual “antithrombotic therapy” and “triple antithrombotic therapy”. It should be rewritten as clear as the end of the introduction of the manuscript. Moreover, the major adverse cardiac events are not defined.

Ans. Thank you for your comment. We have clarified more specifically in the revised abstract regarding the information of dual and triple antithrombotic therapy. Besides, six enrolled studies in this network meta-analysis have their own trial-defined major adverse cardiac events (MACEs), which are not completely the same with each other. The primary efficacy endpoint of our Network Meta-analysis was the trial-defined MACE of each study. Following your suggestion, we have summarized the definition of each trial-defined MACEs in Supplementary Table 1.

  1. It is important for readers to clearly detail what DAPT means in the different studies. Did all the studies consider the same drugs for DAPT?

Ans. Thank you for your comment. All the enrolled patients in WOEST and ISBAR-TRIPLE studies used clopidogrel plus aspirin, wherein PIONEER, RE-DUAL PCI, AGUSTUS, ENTRUST-AF PCI studies, more than 90% of the patients received clopidogrel plus aspirin, only less than 10% received ticagrelor or prasugrel. Following your comment, we stated this information in the result of the main manuscript (line 139-143) and Supplementary Table 1. 

  1. It would have been interesting to perform subgroups analyses, according to age, renal failure, and other comorbidities.

Ans. Thank you for your comment. We agree with you that it would be interesting to have subgroup analyses in this network meta-analysis. However, the subgroup variables, including age, underlining comorbidities, renal function among these 6 studies were different from each other. For example, the cut-off value of the age among these 6 studies subgroup analyses was different. Besides, only ISBAR and AUGUSTUS studies included renal function in their subgroup analysis. Please refer to the table below. 

Variables in subgroup analyses of each studies

Variables

WOEST

ISBAR

PIONEER

REDUAL

AUGUSTUS

ENTRUST

Age

>75 vs. < 75

>74.2 vs. <74.2

>75 vs. <75

>65 vs. <65

>80 vs. <80

<65 vs. 65-80 vs. >80

>65 vs. <65

Sex

V

V

V

V

V

NR

Diabetes

NR

V

NR

V

V

NR

Hypertension

NR

V

NR

NR

NR

V

Heart failure

NR

NR

NR

NR

V

V

Stroke

NR

V

NR

NR

NR

V

Renal function

NR

V

NR

NR

V

NR

Acute coronary syndrome

V

NR

NR

V

NR

NR

Stent type

V

NR

V

V

NR

NR

NR, no reported; V, variables reported in the subgroup analysis

  1. You detailed in the statistical analysis section how you determined the heterogeneity across studied and how you performed sensitivity analyses. However, you did not provide any results. Please clarify this point.

Ans. Thank you for your correction. Heterogeneity across studies was also assessed using the Cochrane Q test and Higgins I2 statistics. The Higgins I2 statistics was used to determine the degree heterogeneity (I2 < 25% - low, 25%–50% - moderate, and > 50% - high degree of heterogeneity) among the enrolled studies. We have included the I2 value in Figure 3 and Figure 4 for each endpoint. We also revised the statistical analysis in the revised main manuscript. Thank you. 

  1. Please add in figure3 for the different endpoints, the individual results of each study you considered, as well as the weight of the different studies included in each analysis and the value of I² for each endpoint.

Ans. Thank you for your correction. We have included the I2 value in Figure 3 and Figure 4 for each endpoint. The weight of each study is essential in Forest plots of meta-analysis but not network meta-analysis, in which each Forest plots included data from several studies.  

  1. There are too many figures. Some of them are not very informative and should be considered as supplemental data only. In particular, the figure 4 is not so easy to understand for readers. Please clarify.

Ans. Thank you for your suggestion. Following your comment, we have revised the description of Figure 4 to improve clarity and we shift Figure 4 to supplementary Figure 2 in the revised supplementary data.

  1. In the discussion, you should more focus on the added value of your results. You should also more discuss your results in the light of the existing literature.

Ans. Thank you for your comment. We have discussed more regarding the comparison between previous and present meta-analyses in the discussion of the revised manuscript (line 261-272). 

  1. There are too many abbreviations in the manuscript making it difficult to read. Same comment for tables.

Ans. Thank you for your comment. Following your comment, we deleted some abbreviations, such as "DAT", "TAT", "NMA" in the revised article. 

  1. Please correct all the spelling errors throughout the manuscript. An English editing by a narrative speaker would be appreciated.

Ans. Thank you for your comment. The revised article was edited by professional English editing service. 

Reviewer 4 Report

  1. English spelling has to be improved. Especially the introduction and Discussion
  2. The bibliography is not addressed in the Vancouver style. 
  3. Figure 4 needs more explanation. Why is distributed in that way?

Author Response

Response to reviewer 4

  1. English spelling has to be improved. Especially the introduction and Discussion

Ans. Thank you for your comment. The revised article was edited by professional English editing service. 

  1. The bibliography is not addressed in the Vancouver style. 

Ans. Thank you for your correction. The bibliography was addressed in the MDPI style according to the author's information. We have revised the bibliography in the revised article. 

  1. Figure 4 needs more explanation. Why is distributed in that way?

Ans. Thank you for your suggestion. Following other reviewer's suggestions, Figure 4 was shifted to supplementary Figure 2. We also have revised the description of this Figure to improve clarity.

Round 2

Reviewer 3 Report

The authors have taken into account all my suggestions and improved their manuscript. My only minor comment is that it would be easier for readers to indicate the I² value closed to each corresponding plot rather than in the legend of the different figures. Best regards. 

Author Response

The authors have taken into account all my suggestions and improved their manuscript. My only minor comment is that it would be easier for readers to indicate the I² value closed to each corresponding plot rather than in the legend of the different figures. Best regards.

Ans. Thank you for your suggestion. Following your comment, we have indicated the I² value in Figure 3 and 4 of the revised article.

Reviewer 4 Report

Has been improved, no comments beyond the corrections

Author Response

Has been improved, no comments beyond the corrections

Ans. Thank you.